# Maltreatment or violence-related injury in children and adolescents admitted to the NHS: comparison of trends in England and Scotland between 2005 and 2011

Arturo Gonzalez-Izquierdo,[1] Mario Cortina-Borja,[1] Jenny Woodman,[1] Jacqueline Mok,[2] Janice McGhee,[3] Julie Taylor,[4] Chloe Parkin,[1] Ruth Gilbert[1]

▶ Prepublication history and additional material is available. To view please visit the journal (http://dx.doi.org/10.1136/bmjopen-2013-004474).

For numbered affiliations see end of article.

**Correspondence to**
Professor Ruth Gilbert;
r.gilbert@ucl.ac.uk

## ABSTRACT

**Objective:** Legislation to safeguard children from maltreatment by carers or violence by others was advanced in England and Scotland around 2004–2005 and resulted in different policies and services. We examined whether subsequent trends in injury admissions to hospital related to maltreatment or violence varied between the two countries.

**Setting and participants:** We analysed rates of all unplanned injury admission to National Health Service (NHS) hospitals in England and Scotland between 2005 and 2011 for children and adolescents aged less than 19 years.

**Outcomes:** We compared incidence trends for maltreatment or violence-related (MVR) injury and adjusted rate differences between 2005 and 2011 using Poisson or negative binomial regression models to adjust for seasonal effects and secular trends in non-MVR injury. Infants, children 1–10 years and adolescents 11–18 years were analysed separately.

**Results:** In 2005, MVR rates were similar in England and Scotland for infants and 1–10-year-olds, but almost twice as high in Scotland for 11–18-year-olds. MVR rates for infants increased by similar amounts in both countries, in line with rising non-MVR rates in England but contrary to declines in Scotland. Among 1–10-year-olds, MVR rates increased in England and declined in Scotland, in line with increasing non-MVR rates in England and declining rates in Scotland. Among 11–18-year-olds, MVR rates declined more steeply in Scotland than in England along with declines in non-MVR trends.

**Conclusions:** Diverging trends in England and Scotland may reflect true changes in the occurrence of MVR injury or differences in the way services recognise and respond to these children, record such injuries or a combination of these factors. Further linkage of data from surveys and services for child maltreatment and violence could help distinguish the impact of policies.

## BACKGROUND

Exposure to maltreatment or violence is common during childhood. The term child

## Strengths and limitations of this study

- Maltreatment or violence-related injuries were analysed only if they resulted in admission to hospital. We could not determine whether children and adolescents with these injuries were instead being managed in other settings such as outpatient or emergency departments or primary care or not presenting to healthcare.
- Differences in coding practices, admission thresholds, performance targets, allocation of resources or service configuration could potentially influence trends.
- Both countries have national health services and standardised, comparable hospital administrative data captured at a national level.
- Analyses took into account background trends in admissions for other types of injuries, thereby partially accounting for admission thresholds related to system factors such as waiting times in the emergency department.

maltreatment refers to any form of physical, emotional or sexual abuse or neglect that results in actual or potential harm to a child.[1] [2] Among young children, maltreatment is mainly caused by carers. During adolescence, maltreatment more often results from violence by people other than carers and includes assault or bullying by peers, siblings, other family members or strangers. Community surveys estimate that 40–60% of adolescents have been exposed to maltreatment by carers or abuse or violence by others during the previous year, and approximately 1 in 10 of them would have been injured as a result.[3] [4] Multiple types of abuse or violence often co-occur and are linked to adverse psychological outcome.[5] [6]

We conducted an ecological comparison of trends in hospital admissions for

maltreatment or violence-related (MVR) injury admissions in England and Scotland from 2005, when both countries implemented new legislation to safeguard children. Scotland also implemented policies from 2005 to reduce violence. We evaluated the continuum of injury related to maltreatment by carers or violence by others (eg, peers, siblings and strangers) for three reasons. First, clinicians dealing with injured children and adolescents may not easily distinguish the perpetrator of an inflicted injury, instead recording uncertainty about the cause or concerns about the child's environment. Second, we measured injury related to maltreatment or violence, rather than definitively caused by, as only a small minority of cases where maltreatment is suspected proceed to child protection assessment and definitive attribution of cause.[7 8] Third, policies to reduce violence may also reduce child maltreatment, and vice versa.[9 10]

Using an ecological comparison of trends in MVR injury admission to hospital, we aimed to generate hypotheses about reasons for variation between the two countries. Correlation with specific policy initiatives is difficult, however, because of the variety of policy, service and societal influences.[11] Policies can impact trends in MVR injury through a variety of mechanisms. Policies to improve recognition of and responses to child maltreatment or violence may increase awareness but could also reduce occurrence. Second, policies affecting socioeconomic inequalities, social cohesion, antisocial behaviour and welfare policies to improve support for disadvantaged families, might also affect rates of maltreatment or violence.[1 12–15] Third, policies that reduce risk factors for serious injury requiring hospital admission, such as use of knives or other weapons, excessive alcohol consumption and unregulated drug use, might reduce the rate of severe injuries requiring admission to hospital.[16 17] We discuss our findings on trends in the two countries in relation to policies to safeguard children and the wider healthcare context.

## METHODS

We analysed trends in monthly population incidence rates of unplanned MVR injury admission to hospital between 1 January 2005 and 31 March 2012 in England and Scotland. We used hospital administrative data for all National Health Service (NHS) admissions of children in England (Hospital Episode Statistics—HES) and Scotland (Scottish Morbidity Records—SMR) to identify unplanned injury admissions using previously published methods (see web table 1 for definitions).[18 19] We defined MVR injury using a cluster of codes from the International Classification of Diseases, 10th Revision (ICD10) recorded in any diagnostic field at discharge (up to 20 diagnostic fields per episode in HES or 6 diagnostic fields in SMR).

Diagnostic coding by professional coders using case notes and discharge letters completed by clinicians is long established and the accuracy has long been debated.[1 20 21] A recent systematic review found moderate accuracy of coding in hospital administrative data in the UK.[22] Studies using internal validation to compare clusters of ICD codes for detecting maltreatment-related injury with case notes or child protection agency data reported high specificity for clinician concerns about maltreatment,[7] and moderate specificity for definitive evidence of maltreatment or child protection agency notification.[23–26] Studies using external validation to determine whether codes in different settings produce similar rates and risk factors provide weak evidence that codes for maltreatment are measuring a similar underlying entity.[11 27]

We used previously developed MVR injury codes that were developed to be consistent with alert features mentioned in the National Institute for Health and Care Excellence (NICE) guidance for considering maltreatment.[11 27 28] An evaluation of this coding cluster against clinical records is reported elsewhere.[7] The cluster of codes includes four subgroups (see web table 1). These comprise specific references to maltreatment syndrome, assault, unexplained injury, based on codes indicating the need for further evidence to determine the intent of injury (undetermined cause), and codes reflecting concerns about the child's social circumstances, family environment and adequacy of care; factors that in combination with an injury should alert clinicians to consider the possibility of maltreatment. We used admission rather than child, as the unit of analysis as very few children (<3%) had repeat MVR injury admissions within a given year (unpublished, data available from authors).

Denominator populations were derived from mid-year population estimates by year of age and calendar year published by the Office for National Statistics in England and the General Register Office for Scotland.[29 30] Analyses were stratified into three age groups reflecting broad stages of dependency, socialisation and exposure to violence (infants <1 year—non-ambulatory, children 1–10 years—ambulatory and mixing socially under parental supervision and adolescents 11–18 completed years—school age and social mixing outside parental supervision), which might be amenable to different policies. MVR injury in infancy is likely to reflect abuse or neglect by carers. Between 1 and 10 years of age, MVR injury can reflect abuse or neglect by carers or inadequate protection of children from abuse or neglect by others. Among 11–18-year-olds, physical violence by carers occurs at least as frequently as at younger ages, but violence due to other family members, peers or strangers becomes much more frequent than that perpetrated by carers.[6 31]

### Analyses

Our study builds on previous reports where we used trends in annual incidence rates in cross country comparisons for children aged 10 years or less.[11 27 28] In this study, we plotted monthly incidence rates using three-monthly moving average rates. We used time series

**Table 1** Hospital admissions for MVR injury in children in England and Scotland between 2005 and 2011, inclusive

| Country | Age group | Total unplanned injury | Unplanned injury rates per 100 000 cy* | MVR injury | MVR incidence rates per 100 000 cy* | Percentage of total MVR injury† | Percentage of total injury‡ |
|---|---|---|---|---|---|---|---|
| England | <1 year | 61 987 | 1361.3 | 3955 | 86.9 | 7.3 | 6.4 |
| | 1–10 years | 402 334 | 962.5 | 7876 | 18.8 | 14.5 | 2.0 |
| | 11–18 years | 414 259 | 1157.4 | 42 376 | 118.4 | 78.2 | 10.2 |
| | Subtotal | 878 580 | 1069.5 | 54 207 | 66.0 | 100 | 6.2 |
| Scotland | <1 year | 5168 | 1277.9 | 290 | 71.7 | 3.9 | 5.6 |
| | 1–10 years | 48 357 | 1244.3 | 570 | 14.7 | 7.7 | 1.2 |
| | 11–18 years | 51 339 | 1471.0 | 6507 | 186.4 | 88.4 | 12.7 |
| | Subtotal | 104 864 | 1347.7 | 7367 | 94.7 | 100 | 7.0 |

*Denominators are the mid-year population estimates from the Office for National Statistics and General Register Office for Scotland.
†Denominator is the total number of MVR injury admissions in children 0–18 years of age by country.
‡Denominator is the total number of unplanned injury admissions within age group and country.
cy, child years; MVR, maltreatment or violence-related.

analyses and fitted segmented Poisson and negative binomial regression models (parametrised as generalised linear models) to determine trends and the possible timing of changes in gradient.[32–35] We took account of underlying trends in injury admission rates by adjusting analyses for unplanned injury admissions that were not related to MVR (non-MVR). We also fitted sine and/or cosine terms to account for annual seasonal variation. Changes in the goodness of fit produced by including these periodic components were measured by the Akaike's information criterion (AIC). To determine whether trends significantly changed direction over time, we fitted segmented models with up to one change point. Negative binomial regression models were fitted to account for overdispersion as the variance of MVR injury rates is likely to increase for increasing rate values. We compared goodness of fit across nested Poisson and negative binomial models using the log-likelihood ratio test. Details of the model are reported in web appendix 1.

Because of the relatively large number of parameters in the models and limited power resulting from small numbers of monthly incidence points, we analysed each country separately and report qualitative differences. In each country, we estimated absolute differences between adjusted rates in calendar years 2005 and 2011 within each age group for MVR and non-MVR injury incidence rates. We also report gradients given by the change point models (and their 95% CIs), and the p values associated with the gradients and, where relevant, with the change in gradient during the study period. A p value <0.05 was considered significant. We plotted smoothed incidence rates predicted by the model, using a non-parametric adaptive smoother.[36] We conducted sensitivity analyses restricted to codes reflecting maltreatment syndrome or assault. Regression models and non-parametric smoothers were fitted using the R environment for statistical computing, V.2.14.2 (http://www.R-project.org).[37] We used the R packages MASS[38] and

SiZer[39] to fit piecewise linear models with Poisson and negative binomial distributions.

## RESULTS

There were 54 207 MVR injury admissions in England and 7367 in Scotland during the study period (January 2005–March 2012). MVR injury admission rates were distributed similarly across age groups in both countries, with the vast majority occurring in adolescents 11–18 years (78.2% in England and 88.3% in Scotland; table 1). The age-related incidence of MVR injury admission was J-shaped with a subsidiary peak in infancy and a major peak in late adolescence (table 1). MVR injury admissions in the 11–18-year-old age group accounted for 10% (England) to 13% (Scotland) of all unplanned injury admissions in this age group.

Figure 1A,B,C shows MVR incidence trends in England and Scotland by age group. Negative binomial models performed significantly better in all age groups (p<0.05); except for the analysis of rates in Scottish infants where the corresponding Poisson model is reported (figure 1 and web table 2). With the exception of infants, monthly trends showed marked seasonal patterns in both countries with peaks in the spring–summer months.

For infants, the incidence of MVR injury admission increased between 2005 and 2011 in both countries: in England 27.8 and in Scotland 23.5/100 000 more infants were admitted for MVR injury in 2011 compared with that in 2005 (table 2, figure 1A). This represented an increase of 37% from 2005 rates in England and of 23% in Scotland (figure 2). The increase in England appears to have been largely determined by the background increase in non-MVR injury admissions as adjustment for the non-MVR trend and seasonal variation showed a 6.6% annual increase that was not significant at the 5% level.

In Scotland, the 18% annual increase in MVR admissions among infants estimated by the multivariable

model did not reach a significant level (p=0.065) and was in contrast to the decline in non-MVR injury admissions (figure 1, web table 2). The absolute increase between 2005 and 2011 in admission rates for non-MVR injury in England and decrease in Scotland is shown in

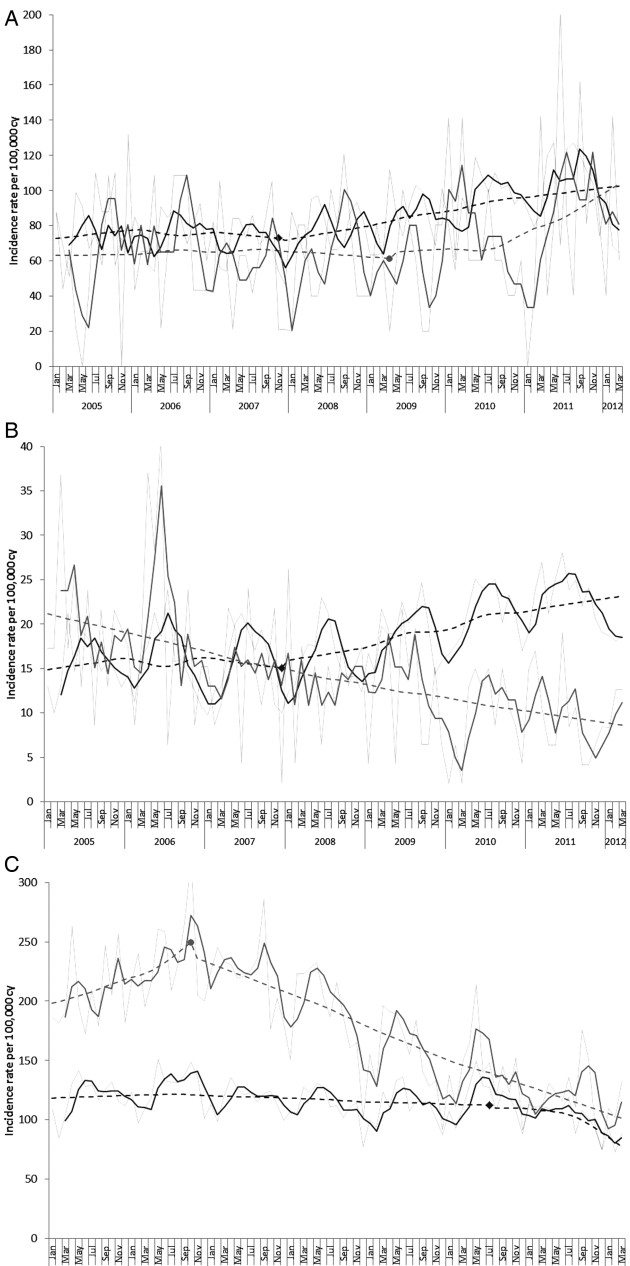

**Figure 1** Monthly incidence trends from January 2005 to March 2012 of maltreatment or violence-related injury in (A) infants, (B) children aged 1–10 years and (C) adolescents aged 11–18 years, in England (dark grey) and Scotland (light grey). Faint lines represent observed rates and bold lines represent three monthly moving averages. Dashed lines represent smoothed trends of incidence rates estimated from the segmented regression analysis (except for trends in Scottish 1–10-year-olds where a standard Poisson regression was used) and markers indicate the change point estimated by the segmented regression model. cy, child years.

table 2 and figure 2. Trends for non-MVR injury are shown in web figure 1A.

Among children aged 1–10 years, the incidence of MVR injury admission increased between 2005 and 2011 in England (rate difference 7.8/100 000) and declined similarly in Scotland (rate difference 9.9/100 000; table 2, figure 1B). This represented an increase of 50% from 2005 rates in England, and a decrease of 50% in Scotland (figure 2). Using multivariable models to adjust for trends in non-MVR injury admissions and seasonal variation, we estimated diverging annual incidence trends: rates were increasing by 10.4%/year in England and declining by 10.8%/year in Scotland. Both these trends were significant at the 5% level (figure 1B, web table 2). In this age group, admissions for non-MVR injury increased in England and decreased in Scotland (table 2 and web figure 1B).

Among adolescents aged 11–18 years, admission rates for MVR injury in 2005 were almost twice as high in Scotland as in England (table 2, figure 1C). A steep decline from the autumn of 2006 in Scotland resulted in converging rates in the two countries by 2011 as rates in England declined more slowly (figure 1C). The absolute difference in rates between 2005 and 2011 resulted in 15.8/100 000 fewer adolescents admitted with MVR injury in 2011 in England and 85.2/100 000 fewer in Scotland, relative reductions of 13.3% and 41.4%, respectively (table 2, figure 2, web table 2). These trends were steeper than the declining trends in non-MVR admissions in both countries, and were significant at the 5% level after adjusting for trends in non-MVR injury admissions and seasonal variation (see web table 2). We estimated an annual decline in the incidence of MVR injury admissions in England of 7.5%, which dated from 2010. The decline for 11–18-year-olds in Scotland was steeper (12.9%) and dated from 2006 (figure 1C, web table 2). The rate of admission for non-MVR injury declined similarly in both countries (figure 2, web figure 1C).

In sensitivity analyses that restricted MVR injury to codes for maltreatment syndrome or assault, qualitative findings were unchanged, but none of the differences between England and Scotland reached significance at the 5% level (see appendix—web table 1). Maltreatment syndrome or assault codes accounted for 63.1% of all childhood MVR admissions in England and 73.5% in Scotland. For infants and 1–10-year-olds, we found weak evidence for increasing trends in England but small numbers in Scotland prevented modelling of trends in these age groups. For 11–18-year-olds, declines using restricted MVR codes were steeper in England (12.4% annually) and similar to the decline in Scotland (12.1%).

## DISCUSSION

Between 2005 and 2011 rates of MVR injury admission increased in England among infants and 1–10-year-olds

**Table 2** Observed mean incidence rate per 100 000 children in calendar years 2005 and 2011 and absolute difference in rates

| Country | Age group | Rate (95% CI) 2005 | | Rate (95% CI) 2011 | | Absolute difference in rates (95% CI) | |
|---|---|---|---|---|---|---|---|
| | | MVR | Non-MVR | MVR | Non-MVR | MVR | Non-MVR |
| England | <1 year | 75.0 (68.1 to 81.9) | 1146.9 (1120.0 to 1173.9) | 102.8 (95.2 to 110.4) | 1334.7 (1307.2 to 1362.2) | 27.8 (27.1 to 28.5) | 187.8 (187.2 to 188.3) |
| | 1–10 years | 15.5 (14.5 to 16.5) | 909.6 (901.9 to 917.3) | 23.3 (22.1 to 24.5) | 950.6 (942.9 to 958.2) | 7.8 (7.6 to 8.0) | 41.0 (41.0 to 40.9) |
| | 11–18 years | 119.1 (116.2 to 122.1) | 1076.5 (1067.6 to 1085.4) | 103.3 (100.6 to 106.1) | 896.2 (888 to 904.5) | −15.8 (−15.6 to −16.0) | −180.3 (−179.6 to −180.9) |
| Scotland | <1 year | 66.1 (44.5 to 87.7) | 1426.3 (1326.0 to 1526.6) | 89.6 (65.5 to 113.7) | 1024.1 (942.5 to 1105.6) | 23.5 (21.0 to 26.0) | −402.3 (−383.5 to −421.0) |
| | 1–10 years | 19.6 (16.0 to 23.3) | 1306.2 (1276.2 to 1336.3) | 9.7 (7.2 to 12.3) | 1136.6 (1108.8 to 1164.3) | −9.9 (−8.8 to −11.0) | −169.7 (−167.4 to −172.0) |
| | 11–18 years | 205.8 (193.4 to 218.2) | 1314.3 (1283.0 to 1345.7) | 120.6 (110.8 to 130.5) | 1113.7 (1083.8 to 1143.7) | −85.2 (−82.6 to −87.7) | −200.6 (−199.2 to −202.0) |

MVR, maltreatment or violence-related.

along with rises in other injury admissions and declined in adolescents, though less steeply than in Scotland. MVR injury admissions in Scotland increased in infants but declined steeply among children aged 1–10 and 11–18 years along with declines in other injury admissions in all age groups. Similarities between England and Scotland were increasing rates of MVR injury admissions among infants and decreasing rates among 11–18-year-olds.

Among 1–10-year-olds, incidence trends for MVR injury admissions diverged between England (increasing) and Scotland (decreasing), but were consistent with trends for other injuries in this age group. Among 11–18-year-olds, rates of MVR injury admission were twice as high in Scotland as in England in 2005, but fell more steeply than in England, resulting in similar rates by 2011.

Limitations of the study centre on whether these trends reflect the occurrence of MVR injury severe enough to require admission or whether they relate to differences in coding or health service thresholds for admission of children with MVR injury. First, one factor contributing to diverging rates could be improvements in the sensitivity of coding in England where coding depth is incentivised by the remuneration system 'payment by results', a system which does not operate in Scotland.[40 41]

Second, changes in admission thresholds could differentially affect rates in both countries. We confined our analyses to admissions, rather than emergency departments (EDs) or primary care because coded data are not available on a national basis for non-admitted patients. However, admissions are the 'tip of the iceberg' in terms of healthcare attendances for MVR injury—reflecting only a minority of those presenting to the ED and primary care.[7 42 43] Flows of patients from the ED to short stay admissions may have increased following introduction of 4 h wait targets in the ED.[44] However, these targets were implemented in Scotland and England in 2004.[45] Moreover, we adjusted trends for background changes in non-MVR injury admissions, which would have been most affected by changes to ED department waiting times.

Differential changes between countries in admission threshold specifically for MVR injuries are possible. We previously reported steep declines in maltreatment-related injury admissions in Manitoba, Canada, following a change in policy to investigate possible maltreatment in the community, avoiding admission to hospital when not medically justified.[11] We are not aware of any explicit policies to shift investigation of alleged maltreatment from the hospital to the community in England or Scotland. However, better coordination of safeguarding services in the community in Scotland compared with England, for example, as a result of the 'Getting it right for every child' (GIRFEC) policy (discussed below), could potentially have contributed to declines in Scotland.

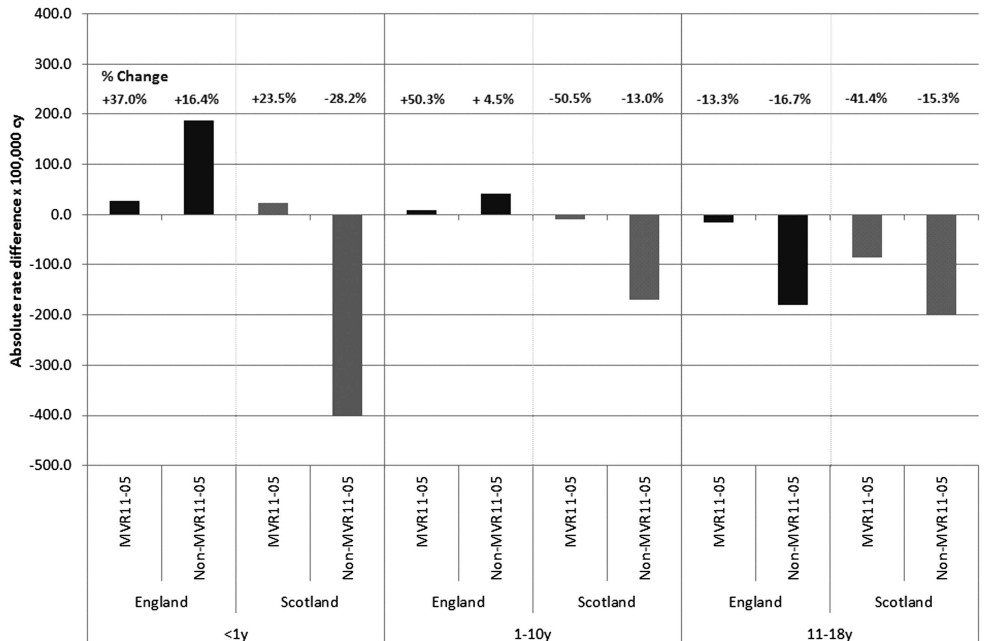

**Figure 2** Absolute rate difference between annual incidence rates in calendar years 2005 and 2011 for MVR injury and non-MVR injury admissions of children and adolescents by age group and country. Note: % Change reflects proportionate change measured as (absolute difference between rate in 2005 and 2011)/(rate in 2005); MVR11–05, difference in maltreatment or violence-related injury admissions between 2005 and 2011 (similarly for non-MVR11–05). cy, child years; MVR, maltreatment or violence-related.

The major limitation of the study is the ecological design, which provides evidence of diverging trends but does not demonstrate which policies or practices might be associated with these different trends.

### Policies related to child maltreatment or violence

To identify policies that may potentially have influenced trends in MVR injury admissions, we asked researchers in England and Scotland to independently list policy initiatives related to child maltreatment that applied nationally in each country, without knowledge of the results of the trend analyses. Relevant legislation, government strategies, inter-agency guidance, implementation or accountability frameworks or health guidance documents were included, but research evidence and guidance by professional bodies were excluded. The results, summarised in web table 3, show at least one set of guidance published in one or both countries every year since 2002.

Both countries saw comparable 'integrative' policies initiated during the first half of the 2000s. 'Every child matters' in England and 'GIRFEC' in Scotland, aimed to promote cooperation across sectors and agencies working for the well-being of children.[46] However, implementation differed between the two countries. In England, 'Every Child Matters' introduced a new structure of children's centres to coordinate targeted and specialist services, separately from existing services.[47] In contrast, GIRFEC promoted coordination within existing structures to integrate universal and targeted services.[48] It is hard to relate the timing of implementation of these two major policy initiatives to changes in trends

observed in our study as implementation was gradual. For example, GIRFEC was launched in 2005 but took several years to be implemented fully.[49 50]

Management of children with suspected maltreatment or neglect within the community may have changed, particularly in Scotland. In both countries, many children with suspected maltreatment are neither admitted to hospital nor attend EDs. However, in Scotland closer working between health and social care professionals as a result of GIRFEC, may have resulted in a large proportion of children receiving medical assessments in outpatient clinics.[51] Many are also being seen by general practitioners (GPs).[43 52 53] Within the NHS, community paediatricians have taken on a major role as 'child abuse paediatricians' and are referred children for assessments in clinics held either within or outside the hospital setting.[51] Hence, small changes in admission thresholds for injury, or for specific subgroups such as those with maltreatment or violence, could have had a substantial impact on rates of admission.

The increase in the incidence of MVR injury admissions among infants in England and Scotland may reflect raised awareness of maltreatment in infants. However, alternative explanations, such as coding practices or a true increase cannot be ruled out. The increase in MVR injury admissions in England predated 2008, when two events occurred that are likely to have raised awareness of maltreatment related injuries presenting to healthcare. The first was NICE guidance 'When to suspect child maltreatment',[54] which was mandated for hospitals in England but not in Scotland. This

guidance coincided with extreme media publicity between October 2008 and August 2009 of the death of Peter Connelly in north London, a 17-month-old child who died from more than 50 injuries inflicted by his parents. These events may have influenced the increase in MVR injury admissions among infants in Scotland, which dated from April 2009, and was in contrast to declines in non-MVR injury admissions among infants.

### Differences in health services

The broader service context may also be relevant. Scotland spends more on the NHS than England, and has more GPs per capita.[55] Approaches to configuration of services post-devolution have been characterised as focussing on markets and management in England and on the medical profession and cooperation in Scotland.[56] In addition, Scotland abolished the purchaser/provider split and the idea of provider competition, and recreated organisations responsible for meeting the needs of the population and running services within defined geographical areas. This may have made it easier to integrate and coordinate services, and therefore improve quality of care along the patient pathway.[55]

### External evidence for changes in trends in child maltreatment and/or violence

Scotland has seen a decline in referrals to the Scottish Children's Reporter Administration over the same period as the decline in MVR injury admissions.[57] Declines in violent crime reported in police statistics have been reported in England and Scotland, and alcohol-related admissions have also declined in Scotland.[58–61] Since 2005, Scotland has implemented intensive programmes to prevent youth violence and reduce drug and alcohol misuse, focussing on vulnerable young people.[62 63] England and Scotland implemented the 'challenge 25' policy in 2009 to reduce youth access to alcohol,[64] but Scotland is planning to introduce minimum pricing for alcohol—a move so far resisted in England (http://www.alcohol-focus-scotland. org.uk/ref).

### Implications

Our analyses show that the incidence of MVR injury admissions in children can change substantially over time and in opposite directions in adjacent countries with similar healthcare systems. The declines in Scotland suggest that the increasing rates observed in England are not inevitable. However, which policies, if any, have influenced these changes cannot be determined from this study. A priority for future research is to distinguish true change in the occurrence of MVR injury needing admission from changes in coding or admission thresholds. This requires analyses of all cases of MVR injury presenting to primary care, those seen as outpatients by community paediatricians, those attending the ED and those admitted to hospital, to understand how children are managed within the healthcare system. Such data linkages are not yet possible due to the lack of well-coded, administrative healthcare databases across health sectors, but are a stated aim of government in England and Scotland.[65 66]

Hospitalisation for maltreatment-related injury or injury due to other forms of victimisation represents considerable suffering to the child and a major cost to the health service. These results strengthen the call by WHO to widen the use of administrative data to improve understanding of how policy can reduce exposure of children to injury due to violence or neglect.[67] Consideration should also be given to linking survey data of adolescent self-reported exposures to health administrative data to measure service use in children and adolescents exposed to maltreatment or violence.

**Author affiliations**
[1]Centre of Paediatric Epidemiology and Biostatistics, UCL Institute of Child Health, London, UK
[2]NHS Lothian University Hospitals Division, Edinburgh, UK
[3]School of Social and Political Science, the Chrystal Macmillan Building, Edinburgh, UK
[4]Child Protection Research Centre, University of Edinburgh, St Leonard's Land, Edinburgh, UK

**Acknowledgements** The authors would like to thank members of the Policy Research Unit for the health of children, young people and families: Terence Stephenson, Catherine Law, Amanda Edwards, Steve Morris, Helen Roberts, Catherine Shaw, Russell Viner and Miranda Wolpert. The authors are grateful to Andrew Woolley for commenting on the policy timeline for England.

**Contributors** RG and AG-I conceived the paper and the analytic plan. AG-I wrote the first draft and carried out the analyses. RG wrote the final version. All authors commented on the analyses and report. RG is the guarantor.

**Funding** This work is supported by awards establishing the Farr Institute of Health Informatics Research at UCLP Partners from the MRC, in partnership with Arthritis Research UK, the British Heart Foundation, Cancer Research UK, the Economic and Social Research Council, the Engineering and Physical Sciences Research Council, the National Institute of Health Research, the National Institute for Social Care and Health Research (Welsh Assembly Government), the Chief Scientist.

**Competing interests** AG-I was supported by funding from the Department of Health Policy Research Programme through funding to the Policy Research Unit in the Health of Children, Young People and Families.

**Provenance and peer review** Not commissioned; externally peer reviewed.

**Data sharing statement** Additional data can be accessed in the web appendix. Source data can be accessed by researchers applying to the Information Services Division Scotland or the Health and Social Care Information Centre for England.

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
