## [Reviewer comments · BMJ Open]

Some articles will have been accepted based in part or entirely on reviews undertaken for other BMJ Group journals. These will be reproduced where possible.

ARTICLE DETAILS

TITLE (PROVISIONAL)	Maltreatment or violence-related injury in children and adolescents admitted to the NHS: comparison of trends in England and Scotland between 2005 and 2011
AUTHORS	Gonzalez-Izquierdo, Arturo; Cortina Borja, Mario; Woodman, Jenny; Mok, Jacqueline; McGhee, Janice; Taylor, Julie; Parkin, Chloe; Gilbert, Ruth

VERSION 1 - REVIEW

REVIEWER	David Finkelhor University of New Hampshire
REVIEW RETURNED	19-Dec-2013

GENERAL COMMENTS	This study uses hospital administrative on admissions to try to describe and understand trends in child maltreatment in England and Scotland. The data bases used in this study are a very promising resource for child maltreatment epidemiologists, particularly in the absence of other large aggregated data sets specifically tailored to child maltreatment. But in general researchers are just in the early stages of using this kind of data for child maltreatment epidemiology, and there are many unanswered questions about the validity of these data for such purposes. So, for example, what kinds of professionals or staff code this data into the data base? What sources are they relying on? How much training have they received in how to code maltreatment? In most new data bases, there is a big learning curve and lots of changes in coding and classifying as the system rolls out: is this data source at that stage of development? Looking over the codes, one certainly wonders if all these components really relate to maltreatment. Some ("problem related to physical environment") certainly sound as if they could apply to accidental as well as neglectful harm. What proportion of what is being counted as maltreatment comes from the more ambiguous as opposed to the more definitive codes? What cautionary experience have researchers in other fields had with using the hospital data? If this were a new instrument to measure CM in survey studies, for example, researchers would be asked to present validity and reliability data as part of their methodology. Presumably someone with training in maltreatment identification would have independently evaluated a set of children to see how many of them were also deemed as maltreated by someone coding from the hospital data available on these same children. This would provide evidence about sensitivity and specificity. The authors cite in the discussion a study of this sort that has been done in Queensland that is encouraging. But it is not at all clear whether such Australian findings apply to England and Scotland, because the history and track record of the data bases may be very different and the ICD
---

	codes do not seem to be identical. Interestingly, the only statement in the paper addressing the validity issue (the codes “have been shown to produce similar trends with age and over time in previous cross-country comparisons”) was confusing to me and did not appear to address many of the key validity issues. It is not clear how cross-country comparison is a validity check for most of the key concerns. Moreover, since the topic of trends is what the current study is intended to address, it would be preferable to use some other criterion for testing validity. The authors do later mention in the limitations section that changes in coding practices or admission thresholds could introduce artifacts or explain the findings. But it would seem important to get into this earlier and generate as many hypotheses about the administrative factors that could explain trends and give them equal billing to hypotheses about policy factors, given that all such explanations are speculative. The goal to evaluate the impact of social policy using trend data is laudable. But there is extremely little evidence in the CM literature about the relationship between policy and administrative data trends. We do not know which kinds of policies are connected with increases or decreases in such data, or whether it is even reasonable to expect someone to be able to detect effects. We also do not know how long such effects might take to manifest. The paper makes a number of assumptions about this, without adequately reviewing whether there is literature to support the conjecture that they could show a relationship. Another issue of concern to this reviewer is the degree to which the findings of this paper duplicate findings previously reported. This research group has published several papers with trend findings for England. Was there actually sufficiently new information in the current report? There was nowhere that the previous trend findings were adequately summarized or the analysis in the current paper justified in the context of the previous findings. Another very important contextual element missing from the paper is some discussion of how and why child maltreatment and violent victimization end up at hospitals. There is research that suggests that only a very small proportion of such victimization is seen by hospitals. An assumption in the paper is that these are the most serious cases or the children with the most bodily harm. But there may be many other reasons for cases to be in a hospital data base, factors that relate to the structure of health care delivery in a region for example, or police investigative practices. For readers to make sense of the meaning of fluctuations in this indicator, it would be very useful to know what are the unique features of hospital screened child maltreatment and whether there is any evidence about regional differences.
--	--

REVIEWER	Koustuv Dalal, PhD Associate Professor (Sr. Health Economist); Chair, International Safe Hospital; Director, Centre for Injury Prevention & safety Promotion (CIPSP)- An Affiliate Support Centre: WHO CC Community Safety Promotion; Editor- in- Chief, WHO CCCSP Safe Community News;
-----------------	--

	Department of Public Health Science School of Health & Medical Sciences Örebro University
REVIEW RETURNED	19-Feb-2014

GENERAL COMMENTS	Introduction: Too short to understand the topic. Rationale of the study is missing. Objective should be stated more clearly. I strongly suggest to expand paragraph 1. Also clearly state the maltreatment and violence. Method: Why infants is <1 year and adolescent from 11? Please provide references for selecting infant, children and adolescent. Statistical analyses should be more specifically described. Results: I strongly recommend to describe the tables and figures more. Discussion: You had an ample opportunity to discuss the excellent results. You can expand it. Please discuss more with the policy issues. Limitations of the study, especially methodological issues should be stated. Recommendations from the study should be there.
---

VERSION 1 – AUTHOR RESPONSE

Reviewer Name Institution and Country New Hampshire	David Finkelhor University of	
1. This study uses hospital administrative on admissions to try to describe and understand trends in child maltreatment in England and Scotland.		No response required
2. The data bases used in this study are a very promising resource for child maltreatment epidemiologists, particularly in the absence of other large aggregated data sets specifically tailored to child maltreatment. But in general researchers are just in the early stages of using this kind of data for child maltreatment epidemiology, and there are many unanswered questions about the validity of these data for such purposes.		We have added a paragraph to the methods to explain that ICD diagnostic codes in hospital administrative data have been used for many years. Accuracy has been debated and we reference a systematic review of studies evaluating the overall accuracy of UK hospital administrative data. We explain that coding is done by professional coders using clinical records. Use of ICD coding for hospital administrative data has been established internationally for decades. ICD coding is coordinated by WHO http://www.who.int/classifications/icd/en/. The validity of ICD coding in hospital administrative data for maltreatment-related injury is discussed in the response to point 6 Text to clarify the above points have been added to the methods.
3. So, for example, what kinds of professionals or staff code this data into the data base? What sources are they relying on? How much training have they received in how to code maltreatment?		Please see response to point 2.  •
4. In most new data bases, there is a big		Please see response to point 2.

learning curve and lots of changes in coding and classifying as the system rolls out: is this data source at that stage of development?	
5. Looking over the codes, one certainly wonders if all these components really relate to maltreatment. Some (“problem related to physical environment”) certainly sound as if they could apply to accidental as well as neglectful harm.	As with all indicators, there is a balance between sensitivity and specificity. We acknowledge in the methods that our cluster of codes does not definitively identify child maltreatment. Our study is not designed to detect only ‘proven’ child maltreatment as previous studies show that only a minority of cases where suspicion of maltreatment was raised about injuries (Flaherty et al 2008) were referred to child protection services. The aim is instead to measure where there are maltreatment-related concerns. As medical records can be requested for viewing by families, inclusion of codes indicating maltreatment-related injury reflects a moderately high threshold of concern. A further factor affecting specificity of coding is that since 2002 coders are required to record diagnostic codes only if the condition is definite or probable. We have previously reported evidence of a shift between using codes reflecting maltreatment syndrome or assault in children less than 5 years old admitted for injury to codes reflecting concerns about the child’s environment or undetermined cause. The overall incidence of any of the four categories of maltreatment-related categories of codes (shown in the appendix table 1) remained the same.(Gonzalez-Izquierdo 2010) This reference has been added to the explanation on coding in the methods. In our response to point 6, we describe studies that have validated the specificity of ICD codes for maltreatment-related injury.
6. What cautionary experience have researchers in other fields had with using the hospital data? If this were a new instrument to measure CM in survey studies, for example, researchers would be asked to present validity and reliability data as part of their methodology. Presumably someone with training in maltreatment identification would have independently evaluated a set of children to see how many of them were also deemed as maltreated by someone coding from the hospital data available on these same children. This would provide evidence about sensitivity and specificity. The authors cite in the discussion a study of this sort that has been done in Queensland that is encouraging. But it is not at all clear whether such	We have added a second paragraph to the methods summarising findings from internal validation and external validation studies for the cluster of maltreatment and violence-related ICD10 codes used in the study.

Australian findings apply to England and Scotland, because the history and track record of the data bases may be very different and the ICD codes do not seem to be identical.	
7. Interestingly, the only statement in the paper addressing the validity issue (the codes “have been shown to produce similar trends with age and over time in previous cross-country comparisons”) was confusing to me and did not appear to address many of the key validity issues. It is not clear how cross-country comparison is a validity check for most of the key concerns. Moreover, since the topic of trends is what the current study is intended to address, it would be preferable to use some other criterion for testing validity.	Please see responses to comment 6
8. The authors do later mention in the limitations section that changes in coding practices or admission thresholds could introduce artefacts or explain the findings. But it would seem important to get into this earlier and generate as many hypotheses about the administrative factors that could explain trends and give them equal billing to hypotheses about policy factors, given that all such explanations are speculative.	Thank you for this suggestion. We have mentioned problems of coding practices in the introduction.
9. The goal to evaluate the impact of social policy using trend data is laudable. But there is extremely little evidence in the CM literature about the relationship between policy and administrative data trends. We do not know which kinds of policies are connected with increases or decreases in such data, or whether it is even reasonable to expect someone to be able to detect effects. We also do not know how long such effects might take to manifest. The paper makes a number of assumptions about this, without adequately reviewing whether there is literature to support the conjecture that they could show a relationship.	We agree that the inter-relationship between policy and the occurrence of child maltreatment needs more research. We also agree that the evidence for an effect of policy on changes in indicators of maltreatment occurrence is limited. We have made these points in the introduction and in paragraph 3 of the discussion
10. Another issue of concern to this reviewer is the degree to which the findings of this paper duplicate findings previously reported. This research group has published several papers with trend findings for England. Was there actually sufficiently new information in the current report? There was nowhere that the previous trend findings were adequately	The focus of this paper is to compare trends in England and Scotland. Modelling of monthly incidence trends, taking into account seasonal variation, has not been previously reported for either country. Annual trends have been reported for England for children under 10 years old (Gilbert, Lancet 2012; Gonzalez-Izquierdo, ADC 2010). We have added a statement to clarify how this work builds on previous reports and referenced the

summarized or the analysis in the current paper justified in the context of the previous findings.	previous reports in the analyses section of the methods, noting that monthly trends and trends across all ages including 11-18 yrs, have not previously been reported.
11. Another very important contextual element missing from the paper is some discussion of how and why child maltreatment and violent victimization end up at hospitals. There is research that suggests that only a very small proportion of such victimization is seen by hospitals. An assumption in the paper is that these are the most serious cases or the children with the most bodily harm. But there may be many other reasons for cases to be in a hospital data base, factors that relate to the structure of health care delivery in a region for example, or police investigative practices. For readers to make sense of the meaning of fluctuations in this indicator, it would be very useful to know what are the unique features of hospital screened child maltreatment and whether there is any evidence about regional differences.	We have added text and references to the discussion to make the point that children admitted to hospital with maltreatment- or violence-related injury are a minority of children exposed who present to health care services. We have restructured and expanded the discussion of where else in the health care system these children may be seen. We have also acknowledged that better coordination of child safeguarding in the community could diminish MVR injury admissions (see discussion).
12. What proportion of what is being counted as maltreatment comes from the more ambiguous as opposed to the more definitive codes?	We have previously reported that approximately 60% of maltreatment or violence related injury admissions were identified by specific maltreatment or assault codes and the remainder by codes reflecting investigation for indeterminate cause of injury or concerns about the child's adverse social circumstances. We have added results of a sensitivity analysis restricting trends to specific codes for maltreatment syndrome or assault. These analyses did not qualitatively change our results but there was limited power to detect an effect in these analyses.
Reviewer Name Koustuv Dalal Institution and Country Koustuv Dalal, PhD Associate Professor (Sr. Health Economist); Chair, International Safe Hospital; Director, Centre for Injury Prevention & safety Promotion (CIPSP)- An Affiliate Support Centre: WHO CC Community Safety Promotion; Editor-in-Chief, WHO CCCSP Safe Community News; Department of Public Health Science School of Health & Medical Sciences Örebro University	
1. Introduction: Too short to understand the topic. Rationale of the study is missing. Objective should be stated more clearly. I strongly suggest to expand paragraph 1. Also clearly state the maltreatment and violence.	We have added an additional paragraph to the introduction to explain the rationale. We have also explained in the introduction that our study includes the continuum of maltreatment related and violence related injury admissions

	across the age range.
Method: Why infants is <1 year and adolescent from 11? Please provide references for selecting infant, children and adolescent.	Infants are defined as children aged less than one year old as stated in the methods. The age groups were based on socio-developmental milestones, which are relevant to the risk of injury and whether injury is likely to be related to parental abuse or neglect or failure to protect or to violence inflicted by people other than parents. The concepts underlying these age groupings are mentioned in:  a) The introduction para 1 b) In detail in the methods – para 2 We have added further text to the methods para 2, to clarify these categories.
Statistical analyses should be more specifically described.	We have added details of the statistical model to the appendix.
Results: I strongly recommend to describe the tables and figures more.	Thanks for this suggestion. We have added a sentence to the results to explain table 1 in more detail.
Discussion: You had an ample opportunity to discuss the excellent results. You can expand it. Please discuss more with the policy issues. Limitations of the study, especially methodological issues should be stated.	We have restructured the discussion to make clearer the different factors potentially contributing to the trends in England and Scotland. Discussion of the limitations has been expanded
Recommendations from the study should be there.	We have added a recommendation for further analyses of indices to determine whether the changes in MVR injury admissions are consistent with changes in indicators of maltreatment or violence in the community measured in surveys.